# Prolonged Low-Dose Administration of FDA-Approved Drugs for Non-Cancer Conditions: A Review of Potential Targets in Cancer Cells

**DOI:** 10.3390/ijms26062720

**Published:** 2025-03-18

**Authors:** Olivia Chang, Sarah Cheon, Nina Semenova, Neelam Azad, Anand Krishnan Iyer, Juan Sebastian Yakisich

**Affiliations:** 1Governor’s School for Science and Technology, Hampton, VA 23666, USA; silentmist03@gmail.com (O.C.); sarahcheon18@gmail.com (S.C.); 2Department of Pharmaceutical Sciences, School of Pharmacy, Hampton University, Hampton, VA 23668, USA; nina.semenova@hamptonu.edu (N.S.); anand.iyer@hamptonu.edu (A.K.I.); 3The Office of the Vice President for Research, Hampton University, Hampton, VA 23668, USA; neelam.azad@hamptonu.edu

**Keywords:** carcinogenesis metastasis, chemoprevention, stem cells, plasticity, senescence

## Abstract

Though not specifically designed for cancer therapy, several FDA-approved drugs such as metformin, aspirin, and simvastatin have an effect in lowering the incidence of cancer. However, there is a great discrepancy between in vitro concentrations needed to eliminate cancer cells and the plasma concentration normally tolerated within the body. At present, there is no universal explanation for this discrepancy and several mechanisms have been proposed including targeting cancer stem cells (CSCs) or cellular senescence. CSCs are cells with the ability of self-renewal and differentiation known to be resistant to chemotherapy. Senescence is a response to damage and stress, characterized by permanent cell-cycle arrest and apoptotic resistance. Although, for both situations, there are few examples where low concentrations of the FDA-approved drugs were the most effective, there is no satisfactory data to support that either CSCs or cellular senescence are the target of these drugs. In this review, we concisely summarize the most used FDA-approved drugs for non-cancer conditions as well as their potential mechanisms of action in lowering cancer incidence. In addition, we propose that prolonged low-dose administration (PLDA) of specific FDA-approved drugs can be useful for effectively preventing metastasis formation in selected patients.

## 1. Introduction

In the United States cancer has been the second leading cause of death behind heart disease for decades [1]. Large bodies of research have been conducted for the prevention of cancer incidence. Of those, there have been reports of certain FDA approved drugs not specifically designed for cancer treatment having an impact on the rate of cancer incidence. For instance, the drug metformin is commonly used to treat type II diabetes for its ability to lower blood sugar levels by limiting the amount of glucose absorbed by the body and increasing sensitivity to insulin [2]. Yet due to demonstrated metformin ability to regulate signaling pathways involved in cell proliferation and apoptosis the drug is regarded as an anticancer agent [3]. When used on diabetic patients for prolonged periods, it decreased the cancer incidence of prostate cancer [4], colorectal cancer [3], and breast cancer [5]. Another example is aspirin which is an analgesic and antipyretic as well as an agent to reduce risk for cardiovascular diseases [6]. Although being FDA approved as a drug for diseases with no previous association to cancer treatment, it was shown to decrease the incidence of colorectal cancer [6], pancreatic cancer [7], and ovarian cancer [8]. One more FDA approved drug that affects cancer incidence is simvastatin. Primarily used as a drug to lower cholesterol, simvastatin reduces the cancer incidence of renal cell carcinoma [9]. Additional effect of simvastatin on apoptosis was demonstrated in endometrial cancer cells [10]. The list of drugs possessing similar effects is already quite lengthy (Table 1) and is expanding as more and more FDA-approved drugs are shown to be effective to reduce the incidence of different types of cancer. Being administered at low doses relatively well tolerated by the human body, such drugs could be widely used in clinical practice to avoid adversary side effects related to the chemotherapy. The goal of this article is to review some of the most used FDA approved drugs prescribed for non-cancer conditions, highlighting the discrepancies between in vivo vs in vitro potency as anticancer drugs and their possible mechanism of action as chemopreventive agents. We will discuss the existing evidence that link these effects on these drugs on Cancer stem cells, cellular senescence, clonogenicity as well as other potential targets.

## 2. Discrepancies Between the Plasma Concentration Found in Patients vs. the In Vitro Concentration Needed to Eliminate Cancer Cells

Plasma concentration, the drug concentration in plasma derived from patient’s blood after drug intake [41], and in vitro concentration, the concentration typically used in in vitro assays [42], are the critical factors to evaluate the efficiency of each type of drug. Discrepancies between the plasma concentration vs. the in vitro concentrations needed to eliminate cancer cells have been described for each type of drug. In general, the in vitro concentrations required to reduce the number of cancer cells are of a higher value than the plasma concentrations in the body [42]. Metformin, for example, has displayed a plasma concentration of 0.00116 mM (<1.5 μg/mL) [43] while its in vitro concentration that inhibits the proliferation of cancer cells was typically 5–30 mM [44]. As such, the difference between the in vivo and in vitro concentrations is more than a 4310 fold. Furthermore, a low concentration of metformin (0.2 mM) has a selective effect on the growth of pancreatic adenocarcinoma AsPC-1 and SW1990 cells, based on the differential expression of the surface markers [44]. After treatment with 0.2 mM metformin in vitro, $\mathrm{the}\;\mathrm{proportion}\;\mathrm{of}\;CD133^{+}$ cells was reduced by inhibiting proliferation through G_1_/S arrest, but not by apoptosis. However, low concentrations of metformin did not affect $\;CD24^{+}$,$\;CD44^{+}$, $ESA^{+},\;$ or $CD24^{+}CD44^{+}ESA^{+}$ cells. It is interesting, nevertheless, that the low concentrations of metformin reduced the invasion of pancreatic cancer cells in vitro as well as inhibited the pancreatic cancer xenograft growth in vivo (the plasma concentration of 0.02 mm) [44]. For comparison, the peak plasma concentration for aspirin was found to be up to 0.304 mM (54.25 mg/L) [45] and the IC_50_ obtained in vitro for 72 h treatment was 5 mM for MDA-MB-231 and 2.5 mM for MCF-7 breast cancer cell lines [46]. The comparison between plasma and the in vitro concentrations demonstrates more than a 16-fold difference. The similar difference can be observed in the case of simvastatin which has displayed peak plasma concentrations ranging from 0.08 to 2.2 and from 0.03 to 0.6 μM for simvastatin lactone and carboxylate, respectively [47], while its in vitro concentration was revealed to be between 0.001 mM and 0.005 mM, creating a >10-fold difference. Treatment of MDA-MB-231 breast cancer cells with a 1–5 µM concentration of the drug was sufficient to increase the expression of tumor-suppressing genes p21 and p27 as well as the expression of miR-140-5p, which plays the role of tumor-suppressor in breast cancer, resulting in induced apoptosis and inhibited cell proliferation [13]. Therefore, it can be stated that the effectiveness of the drugs on cancer demonstrates a great disparity between the plasma concentration and in vitro concentration. Other examples can be referred to in Table 2.

The discrepancies cannot be explained by pharmacokinetics factors present in in vivo but absent in in vitro experiments. For instance, in in vitro experiments, there are no metabolic elimination and clearance of drugs by other organs such as liver and kidneys. Thus, it would be expected that if pharmacokinetics factors play a role, these drugs will be more potent in vitro than in vivo. Moreover, while the cell lines used in vitro experiments are mainly from commercial sources that might be different from the cancer cells in vivo this still does not explain the general discrepancy. This assumption is supported by the fact that when drugs are tested in commercial cell lines and patient-derived cell lines, the overall potency is similar. For instance, we found that commercial DBTRG.05MG glioma cells showed similar sensitivity to menadione and vitamin C compared to a panel of eight different glioma patient-derived cell lines [48].

**Table 2 ijms-26-02720-t002:** Typical in vitro versus in vivo concentrations of selected FDA-approved drugs (except genistein and resveratrol, not FDA approved) prescribed for non-cancer conditions for extended periods of time. (N/A = cannot be calculated due to lack of in vitro data).

Drug	Typical Duration of Treatment	Typical Plasma/Serum Concentration	Typical In Vitro Concentration	Lowest In Vitro/Typical Plasma Concentration **	References
Metformin	Long Life	<1.5 μg/mL (0.00116 mM)	5–30 mM	4310.345	[43,44]
Glimepiride	Long life	326.6 ± 98.5 μg/L (0.000465–0.000866 mM)	No information in vitro	N/A	[49]
Glimepiride-metformin	Long life	168.2 ng/mL (0.00040295 mM)	0.025–0.4 mM	62.042	[15,50]
Aspirin	Many months and years	0.29–0.66 mg/L (0.00161–0.00366 mM)	1.0 mM and 5.0 mM	273.224	[45,46]
Salicyluric acid (aspirin metabolite)	Same as aspirin		No significant effect on cell proliferation		[51]
Salicylic acid (aspirin metabolite)	Same as aspirin	0.23–25.40 µM	6 mM	236–26,000	[52,53]
Gentisic acid (aspirin metabolite)	Same as aspirin	5–25 µM	14 mM	560–2800	[54,55]
Statin	Long life	1.6–15 nmol/L (0.0000016–0.000015 mM)	10–160 μM (0.01–0.16 mM)	666.667	[56,57,58]
Simvastatin	Long life	9.02 ± 1.18 ng/mL (0.0000216 mM)	1–5 µM (0.001–0.005 mM)	46.296	[59]
Glipizide	Long life	Varying between 380 and 611 ng/mL (0.85–1.35 nmol/mL) OR 0.000853–0.00137 mM Plasma drug concentration declines to 12.0 ng/mL after 24 h	25–100 μM (0.025–0.1 mM)	18.248	[60,61]
Empagliflozin	Long Life	25–600 ng/mL (0.0000554 mM) (0.00133 mM)	50 μM (0.05 mM)	37.594	[62,63]
Naproxen	Long Life	35 ± 0.4 micrograms/mL (0.152 mM)	1–10 mM	6.579	[17,18,64]
Etoricoxib	Maximum 8 days	1206.4 ng/mL (0.00336 mM)	No information in vitro.	N/A	[65]
Everolimus	As long as it is working or until there are side effects	15.3 ng/mL (0.0000160 mM)	0.1 μg/mL (0.000104 mM)	6.5	[66,67]
Exemestane	Five to ten years	22.1 pg/mL (0.0000000746 mM)	0–50 μM (0–0.05 mM)	670,241.287	[68,69]
Goserelin (Zoladez)	Long-term unless clinically inappropriate	8 ng/mL within the first 24 h with 10.8 mL depot (0.00000630 mM)	$10^{-9}-10^{-3}$ mol/L (0.000001–1 mM)	0.159	[70,71]
Raloxifene	Long-term treatment (more than 3 years)	0.5 ng/mL (0.00000106 mM)	$10^{-9}$–10^−7^ M (0.000001–0.00001 mM)	0.943	[72,73,74]
Lenalidomide	Multiple lines of therapy until stalled disease progression or death	0.92 and 2.447 μg/mL for PO (oral) administration (0.00355 mM and 0.00245 mM)	100 μM (0.1 mM)	40.816	[75,76,77]
Phenformin	12 weeks at most	241 ng/mL (0.00117 mM)	0.01–10 micrograms/mL (0.0000487–0.0487 mM)	0.0416	[78,79]
Degarelix (Firmagon)	Long-term (after 7 months of degarelix, patients with PSA less than 4 ng/mL stop therapy until PSA rises to over 4 ng/mL (maximum 24 months)).	5–50 nM (0.000005–0.00005 mM)	10 μM (0.01 mM)	200	[80,81]
Resveratrol	Short (<6 months); Medium (6 months to 2 years; Long (>2 years)	539 ± 384 ng/mL (0.00236 ± 0.00168 mM)	25 μM (0.025 mM)	10.593	[82,83,84,85]
Genistein	Long-term	14 ng/mL (0.0000518 mM) * Note–Western Population	~150 μmol/L (0.15 mM)	2895.753	[86,87,88]

* No data were found in vitro. ** Lowest in vitro/Highest in plasma = most useful concentration for elimination of cancer cells for prolonged time.

## 3. Administration of Classical Anticancer Drugs for Cancer Conditions

In general, for classical anticancer drugs commonly used for cancer treatment, there are no discrepancies between the plasma concentration found in patients vs. the in vitro concentrations needed to eliminate cancer cells (Table 3). Doxorubicin (DOX), a chemotherapeutic agent frequently used for the treatment of a variety of cancers, provides a good example. The mechanism of its cytotoxic action is multiple including DNA intercalation and adduct formation, topoisomerase II (TopII) poisoning, the generation of free radicals and oxidative stress, and membrane damage through altered sphingolipid metabolism [89]. The plasma concentration found in patients ranges between 0.023 and 1.14 μM [90] and the IC50 for of DOX in MDA-MB-231, MCF-7, MDA-MB-468, and 4T1 was 0.28 µM, 0.14 µM, 0.13 µM and 0.11 µM, respectively [91]. Thus, the plasma concentration of DOX achieved in plasma is high enough to eliminate several types of cancer cells in vitro. The failure of DOX to eliminate cancer cells in vitro is attributed to resistance with some cell lines exhibiting a resistance index >100 [92]. Another example is cyclophosphamide and its main active metabolite phosphoramide mustard. The cyclophosphamide serum level can reach up to 175 µM, a concentration higher than the in vitro IC50 described for human HL60 cells (IC50 = 8.79 μM) [93] or mouse BALB/c 3T3 cell (cells (IC50 = 37.6 μM) [94]. Peak plasma levels of phosphoramide mustard of 50 to 100 μM were found at 3 h after cyclophosphamide administration [95], a concentration range higher than the in vitro IC50 for several cancer cell lines including V-79 Chinese hamster lung fibroblasts (IC50 = 1.8–69.1 μM) [96] and rat spontaneously immortalized granulosa cells (IC50 = 3–6 μM) [97]. At these concentration ranges (3–6 μM), phosphoramide mustard induces DNA adduct formation and ovarian DNA damage and increases DNA damage responses (DDR) gene mRNA expression levels and DDR protein within 24–38 h [97].

Capecitabine is a chemotherapeutic drug for the treatment of patients with metastatic breast cancer, metastatic colorectal cancer, pancreatic adenocarcinoma, and gastrointestinal cancer. Capecitabine is a prodrug effective when it is metabolized to 5-fluorouracil (5-FU) through three enzymatic reactions [98]. After oral administration, both capecitabine and 5-FU reach the peak concentration in the plasma within 2 h and their elimination half-life is less than 1 h [99]. However, prolonged oral administration of capecitabine is shown to increase the elimination half-life up to 11 h [100]. Peak plasma levels of 5-FU detected in patients range from 0.845 µM 1 h after capecitabine administration [98] to 2 µM [101] and up to 31 µM [99] 2 h after the drug administration. Interestingly, the IC50 for 5-FU measured in different cancer cell lines falls into an interval between 0.2 and 55 µM [102,103,104,105,106]. Since the accepted protocol for capecitabine treatment requires oral administration twice a day for 14 days every 3-week cycle [107,108,109], plasma levels of 5-FU are maintained in human patients comparable to its IC50.

**Table 3 ijms-26-02720-t003:** Typical in vitro versus in vivo concentrations of selected FDA approved anticancer drugs prescribed for cancer conditions.

Drug	Typical Duration of Treatment	Typical Plasma/Serum Concentration	Typical In Vitro Concentration	Lowest In Vitro Effectiveness Within Plasma Concentration Range
Doxorubicin	21-day or 29-day cycle [110]	52.5 ng/mL [111] 12.54–620.01 ng/mL = 0.023–1.14 μM [90]	60 ng/mL for MCF-7 [112] 0.28 µM, 0.14 µM, 0.13 µM and 0.11 µM for MDA-MB-231, MCF-7, MDA-MB-468, and 4T1, respectively [91]	YES
Cyclophosphamide monohydrate	Standard tretament: Up to 90 days [113]	48.97 μg/mL = 175 μM [114]	8.79 μM human HL60 cells [93] 37.6 μM BALB/c 3T3 cells [94,115]	YES
Phosphoramide mustard (Cyclophospahmide metabolite)	Same as cyclophosphamide	50 to 100 μM [95]	1.8–69.1 microM (V-79 Chinese hamster lung fibroblasts) [96] 3–6 μM reduces cell viability in rat spontaneously immortalized granulosa cells (SIGCs), induces DNA adduct formation and ovarian DNA damage and increases DNA damage responses (DDR) gene expression levels and DDR protein [97]	YES
Capecitabine	14 days every 3-week cycle [107,108,109]	9.16 µM [116]	860 μM to 6000 μM [117]	NO (prodrug)
5-fluorouracil (5-FU)	Same as capecitabine	8.845 µM [98]	0.2–55 μM [102,103,104,105,106]	YES
Etoposide	Standard treatment: Five days in a 21–28 days cycle [118] Up to 11 weeks [119]	0.6 to 2.5 micrograms/mL = 1.02–4.25 μM [120]	4.02 ± 4.07 μM (range: 0.242–15.2 μM)) for a panel of 35 etoposide-sensitive cell lines [121]	YES

## 4. Cancer Stem Cells as Targets of PLDA

Stem cells (SCs) are cells that could develop into many different cell types. They also have capacity for self-renewal, which generates more undifferentiated stem cells, while the differentiation gives rise to mature cell types. A small subpopulation of cells within tumors, which demonstrates characteristics of both SCs and cancer cells, is named cancer stem cells (CSCs) [122]. A notable feature of CSCs is their ability to start tumors when transferred into an animal host even in as small amount as 100 cells [123]. CSCs are also characterized by the expression of cell surface markers, which are utilized to isolate and enhance CSCs. Interestingly, the expression of such markers is tumor subtype specific: CD44^+^CD^−/low^ lineage and ALDH+ are abundant in breast CSCs [124,125], CD133^+^ for colon [126], brain [127] and pancreas [128], CD44^+^ for head and neck [129] and cervix [130], CD90^+^ for liver CSCs [131] and head and neck [132] cancers. CD133^+^ has been utilized to identify a radioresistant subpopulation of glioma cells, demonstrating that radioresistance increased DNA repair in glioblastoma CSCs and pointing to the expression of CD133 as a prediction factor of clinical outcomes for patients with glioma [133]. CSCs are also shown to play an important role in developing chemotherapy resistance in different types of cancer [123,134,135]. In addition, CSCs have a significant impact on cancer relapses and metastasis [136,137,138]. Multiple subpopulations of CSCs have been detected in different types of tumors, providing a concept of tumor heterogeneity, which refers to the biological differences between malignant cells of the same tumor arising from genetic and nongenetic mechanisms, which are responsible for the degree of resistance of cancer cells to a certain anticancer drug. This concept was further developed in the classical “Cancer Stem Cell Theory” (CSCT), postulating the hierarchical organization where a subset of CSCs can irreversibly differentiate into all types of non-CSCs. Thus, it should be sufficient to eliminate only the rare subpopulations of CSCs to effectively heal a cancer patient or at least reach a significant improvement [139,140,141]. According to the CSCT, tumor heterogeneity results from the division of cancer stem cells producing cells with differing states of differentiation or stemness [142]. Nevertheless, this concept was found insufficient to explain the experimental findings using only the hierarchical rigid model [143,144,145]. An alternative plasticity model, the “Dynamic CSC Model” (DCSCM), put forward the idea that differentiated tumor cells and cancer stem cells can interconvert into each other [146]. Consequently, each cancer cell has the potential to obtain a cancer stem cell phenotype. Another model, the “Stemness Phenotype Model” (SPM) explains inconsistencies observed with experimental data not suitable to the CEM or the CCSCM. This model illustrates a non-cancer cell evolving into a cancer cell that divides symmetrically during carcinogenesis. Through a process known as interconversion, any cancer cell can acquire a different phenotype depending on the microenvironment, and, following substantial changes in the microenvironment conditions, phenotypic changes are viable in the surviving cells [147]. An additional model, the complex system model (CSM), implies that genetic and epigenetic transformations might occur within a single tumor, developing a multifaceted cell system consisting of coexisting tumor-initiating cell types. The intervention of the cell-cell and cell-niche interactions may weaken the entire tumor system, while every potential tumor forming cells must be targeted for effective therapy according to this model [148]. These alternative models of cancer biology strongly suggest that simply targeting CSCs will not be sufficient to either eradicate cancer or prevent carcinogenesis.

## 5. Cellular Senescence as a Target of PLDA

In non-cancer cells, senescence is an irreversible response to damage that may occur to cells with age or cells undergoing prolonged stress [149]. Senescence can be naturally caused by the shortening of telomeres, the protective chromosomal termini [150]. Telomeres shorten with every cell division due to DNA polymerase inability to completely replicate the lagging strands. When telomere length reaches a critical point, their protective structure is disrupted, leading to telomere disfunctions including the chromosomal fusion. To prevent such outcomes, cells undergo a transition to a non-dividing state, which limits the expansion of undesired cell population [150]. Other cellular conditions that can lead to senescence development include oncogene activation, oxidative stress, mitochondrial dysfunction, irradiation, and exposure to chemotherapeutics [151]. Therapy-induced senescence (TIS) nowadays is a well-established result of conventional cancer therapy. The primary cause of TIS is DNA damage [151,152], which initiates DDR by p53-facilitated translation of p21, a cyclin-dependent kinase inhibitor that prevents cell-cycle progression [153,154,155]. The next phase, senescence maintenance, has been shown to be based on p16 activity [156], which prevents phosphorylation of retinoblastoma protein (Rb) family members and promotes the formation of Rb/E2F complex, facilitating chromatin alterations, mainly histone 3 lysine 9 trimethylation (H3K9me3), which was considered to permanently arrest the cells in the G1 phase [157,158,159]. TIS was considered a favorable outcome of the therapy, as growth arrest and DNA damage associated with senescence have been shown to prevent uncontrollable cell proliferation and eliminate cancerogenic mutations from being passed to the next generations of cells [160]. Nevertheless, further studies demonstrated that at least a subpopulation of tumor cells can escape TIS [161] and give rise to a more aggressive cancer phenotype able to overcome the cell-cycle blockade [162]. An important characteristic of senescent cells is their high resistance to apoptosis [163,164,165]. Thus, senescent state allows the cancer cells to avoid therapy-induced apoptosis [166]. Later, malignant cells can escape senescence, consequently re-entering the cell cycle and causing tumor recurrence [167,168,169].

In the senescence maintenance phase, senescent cells acquire senescence-associated secretory phenotype (SASP), which can modulate signaling pathways in neighboring cells and tissues through secretion of cytokines, chemokines, growth factors and mRNAs, mostly in extracellular vesicles. SASP can also promote tumorigenesis by creating inflammatory microenvironment through the enhanced expression of cytokines and chemokines [170,171,172], especially IL-6 and IL-8, which lead to increased blood supply and tissue repair [173,174], thus supporting tumor progression, invasion and metastasis [175,176]. Other components of SASP include matrix metalloproteases which create tumor-favorable microenvironments [177,178], VEGF to promote angiogenesis [179], as well as factors promoting epithelial–mesenchymal transition [180] and inducing cancer stem cell-like phenotype [181]. On top of that, SASP in chemotherapy-treated cancer cells can produce highly chemotherapy-resistant cell populations [182,183]. Hence, senescent cells can implement both beneficial and adverse effects on tumor progression, which make the senescent cells a very important target for cancer therapy.

## 6. Potential Mechanism of Action at Low Doses

Several FDA-approved drugs not designed for cancer-related application exhibit pharmacological properties which can be beneficial for cancer therapy. For example, aspirin is a potent inhibitor of NF-κB [184], suggesting that this drug can contribute to the elimination of CSCs. Indeed, daily aspirin use has been shown to reduce the risk of colorectal [185], pancreatic [186] and esophageal and gastric cancer [187], recurrence of breast cancer [188], as well as to reduce death due to several common cancers [189]. In vitro experiments using aspirin concentrations equate to the plasma levels between 1 and 5 mM demonstrated a decrease in CSCs markers expression (c-Met, CD44, Ki67, CxCR4); and inhibition of ALDH1 activity and spheroid formation in pancreatic adenocarcinoma AsPC-1 cells. Furthermore, xenograft pancreatic tissue from mice treated with aspirin revealed a reduction in SOX2, CD133, p65 and TNF-α, as well as the ECM components fibronectin and collagen. A recent study found that aspirin decreases metastasis in a mouse model by increasing T cell activation at the metastatic site, provoking immune-mediated rejection of lung and liver metastases [190]. These findings suggest that aspirin targets highly aggressive cancer as well as non-cancer cells. In comparison, low concentrations of metformin did not inhibit proliferation of pancreatic cancer cells but decreased the proportion of $CD133^{+}$ cells, a type of pancreatic CSCs, in a dose-dependent manner through specifically inhibiting their proliferation by G1/S arrest, after the cells were treated with 0.1–0.2 mM of metformin for 72 h. Moreover, xenograft experiments proved the effect of low-dose metformin on pancreatic cancer in vivo, as oral administration of metformin significantly inhibited xenograft growth. The observed effects are attributed to the inhibitory activity of metformin on Erk and mTOR in CD133^+^ cells [44].

Alternatively, metformin has been shown to inhibit the SASP by inhibition of the NF-κB pathway, ultimately limiting the expression of inflammatory cytokines. Indeed, at doses of 1 mM or higher, metformin reduced cytokine gene expression for senescent cells, but did not affect cell proliferation, while as the doses were reduced to 0.5 mM, it became moderately stimulatory for proinflammatory cytokines such as IL6 and IL8, but inhibitory for CXCL5. Therefore, high doses of metformin can impede the negative effects of senescent cells without compromising its anticancer effects [149].

Salinomycin, a drug that inhibits the proliferation of cancer stem cells, has also been shown to significantly reduce the number of senescent glioma cells in vitro. Glioma is a difficult tumor to treat, often involving combined treatment such as surgery, radiotherapy, and chemotherapy. Upon treatment of glioma cells with a high concentration of hydroxyurea (HU) or aphidicolin, a fraction of the cells survived and further on began a cycle of re-growth. Surviving cells displayed senescence-associated-β-galactosidase staining, as well as arrested cell division and flat morphology, which are characteristic features of senescent cells. When these cells were then treated with a low dose (0.5 μM) of salinomycin for 72 h, surviving cells were not detected, and re-growth was prevented. The treatment with even a lower concentration (0.25 μM) did not kill the surviving cells but prevented the re-growth. This two-step treatment not only opens up doors for a safer way to treat the tumor without high toxicity for the patient, but provide the principle which can be applied to other senescent cancer cells [191]. Indeed, recent studies have validated such an approach to anticancer therapy [192,193,194].

Taken together, these data suggest the existence of multiple mechanisms through which PDLA of some drugs may target cancer cells (stemness senescence, clonogenicity), but still the evidence is scarce to conclude that these mechanisms are the main target of PLDA of FDA-approved drugs.

## 7. Alternative Targets: Clonogenicity and Cellular Plasticity

The fact that there are limited data supporting that PLDA of FDA-approved drugs can eliminate CSCs or senescent cells suggests that other cellular processes may be important and are worth considering. We suggest that clonogenicity as well as cellular plasticity may play a role. While this suggestion is merely speculative, due to the lack of available experimental data, they could be explored in future experiments. Recent study performed in our lab pointed to clonogenicity as one more potential target of PDLA. We compared the effect of either nigericin (antibiotic active against gram positive bacteria) or menadione (vitamin K3) on viability and clonogenicity of lung carcinoma A549 and H460 cell lines as well as breast carcinoma MCF-7 and MDA-MB-231 cell lines. The ability of either drug to eliminate cancer cells was 2–10-fold more potent in the colony forming assay than in the viability assay, suggesting that PDLA of certain drugs targets clonogenic rather than proliferation pathways. Our data also revealed the existence of short post-reattachment window of time when cancer cells growing at low density are more sensitive to specific drugs [195]. Thus, PDLA of such drugs can eliminate cancer cells when they are highly sensitive immediately after reattachment, preventing in this way the formation of metastasis.

Clonogenicity is the ability of a single cell to proliferate and develop into a full tumor. The clonogenic assay is an in vitro cell survival assay formed on the ability of one cell to grow into a colony. It has been used as a measure of CSC stemness, the cells’ potential for proliferation without bounds, self-renewal, differentiation into multiple tissue types within a lineage, and tumorigenicity [196]. Through these clonogenic assays, it can be demonstrated that a single CSC can generate clonogenic colonies, supporting its potency for cancer metastasis and repopulation after treatment [196]. In previous studies, both holoclones and meroclones of prostate cancer cell line DU145 were shown to contain cells having stem cell qualities, based on the analysis of the colony-forming ability, transplantation capacity and marker expression. In addition, the presence of CSCs in different type of colonies was confirmed by positive stem cell markers (CD44, α2β1 integrin, Oct4 and BMI1) staining [197]. Another essential characteristic of tumors is intratumor heterogeneity, the biological difference that exists amongst malignant cells from the same tumor, which is responsible for the impaired response against particular anticancer drugs [147]. Recent studies illustrate the complex interplay between clonogenicity, stemness, and intratumoral heterogeneity [198,199,200,201,202]. Unfortunately, for technical reasons, most of the experiments using clonogenic assays have been performed for short periods (3–10 days) with drug concentrations typically used for in vitro experiments (high) and not with concentrations typically found in the plasma of patients.

Cancer cell plasticity refers to the ability of the cell to reversibly shift (interconversion) from a differentiated state with limited tumorigenic abilities to an undifferentiated cancer stem cell state that promotes rampant cell division and tumor growth. Stem cell plasticity represents one of the major therapeutic challenges for differentiation therapies. Poorly differentiated tumors (portraying a mesenchymal phenotype) are better suited to tolerate chemotherapy, while well-differentiated tumors are more sensitive to treatment [203,204]. Plasticity can also dictate the changes between distinct CSC states such as ones with varying specialization for invasion and metastasis [205]. Cancer cell plasticity has been linked to the epithelial–mesenchymal transition program, which describes a constant shift through the spectrum of phenotypic states, capable of driving local invasion, generate cancer stem cells and facilitate metastasis by the dissemination of circulating tumor cells [206]. By blocking interconversion, it may be possible to prevent tumorigenesis and metastasis, but at present, the underlaying mechanism of interconversion is poorly understood and there are no data of drugs affecting interconversion at concentrations found in patients.

## 8. Other Potential Targets of PLDA for Specific Drugs

FDA-approved drugs such as metformin, aspirin, and simvastatin are being presented as alternative methods for chemoprevention as they are capable of eliminating cancer cells in in vitro experiments. However, the in vitro concentrations needed to affect these cancer cells are often more than a 1000-fold higher than the plasma concentration supported by the body. Even in the case of CSCs and senescence where low doses of the FDA-approved drugs have been found to affect the proliferation of cells, the in vitro concentration remains too high for the body to sustain. The fact that PLDA of these drugs indeed decreased the incidence of certain types of cancers indicated that there are essential cellular processes not yet identified that can be the actual target of these drugs. In this regard, genistein may offer some clues. Genistein is a protein tyrosine kinase and topoisomerase II inhibitor present in soy, that decreases the incidence of breast, colon, and prostate cancers. The disparity between the plasma concentration and the in vitro concentration needed to eliminate cancer cells is high. The plasma concentrations of genistein in the European and the Asian population were compared. The median circulating genistein concentration in the top fifth of the distribution amounted to 14 ng/mL in the European population (39) being 7-fold lower than the 99 ng/mL found in Japanese men [207], showing the varying degrees of plasma concentrations based on the diet of the population studied. However, the concentration needed to reduce 50% of the autophosphorylation of the EGF-R-associated tyrosine kinase is 2.7 μM, similar to the 2.4 μM plasma concentration found in Asians with a traditional diet with high soy product consumption [40]. This raises the possibility that there may be other more specific targets, such as the inhibition of autophosphorylation of the EGF-R-associated tyrosine kinase, that may be the target of genistein when present for prolonged periods at concentrations found in vivo.

Additionally, resveratrol, a phytochemical that targets cancer stem cells, has been effectively used in traditional medicine for over 2000 years [82]. The drug possesses anti-oxidant, anti-inflammatory, cardioprotective, and anticancer properties [82]. Resveratrol can reverse multidrug resistance in cancer cells, and, when used in combination with clinically used drugs, it can sensitize cancer cells to standard chemotherapeutic agents or radiation [208,209,210,211]. Multiple effects of resveratrol on cancer include reducing oxidative stress [212,213,214], arresting cell cycle and promoting apoptosis [215,216,217], decreasing inflammation-related tumorigenesis through inhibition of STAT3 [218,219], and modifying tumor microenvironment to reduce its progression and invasion [220,221,222]. However, cancer patients did not sustain the amount of resveratrol plasma concentration compared to the drug concentration required to eliminate cancer cells in vitro, suggesting the ineffectiveness of this drug in decreasing the incidence of cancer, although some positive effects of resveratrol on colorectal cancer were revealed in clinical trials [223,224,225]. Still, as in the case of genistein, the multiple effects of resveratrol suggest that the lower incidence of some types of cancers could be the effect of very specific cell type-dependent processes.

## 9. Concluding Remarks

Most FDA-approved drugs that prevent the incidence of cancer require a much higher concentration in vitro than in vivo to effectively eliminate cancer cells. In special circumstances, low doses of these FDA-approved drugs, including metformin, aspirin, and simvastatin, have been shown to inhibit cancer cell growth by specifically targeting CSCs and senescent cells. Thus far, specific mechanisms of targeting CSCs or senescence pathways through chemoprevention therapy have not been identified, so the data related to those processes are insufficient. However, the potential to discover these mechanisms and apply them to the administration of various FDA-approved drugs opens possibilities to the improvement of current chemoprevention techniques.

On the other hand, the available data shown in Table 1 indicate that there is no universal drug that can lower the incidence of all types of cancer. This notion is consistent with the paradigm that all existing anticancer drugs are cancer type specific, targeting different mechanisms involved in the development and progression of tumors. Hence, each type of cancer may be prevented only by a few specific drugs. For instance, metformin reduces the incidence of prostate, colorectal and breast but there is no known effect on, for example, renal cell carcinoma or pancreatic cancer. This seems to be the trend for all FDA-approved drugs known to reduce cancer incidence. It is also important to clarify that metformin does not prevent 100% but only reduces the incidence of breast cancer. Thus, even for a particular type of cancer, only a fraction of patients will benefit by taking metformin, so the likelihood of eliminating cancer incidence with a single “magic” chemopreventive agent is remote. However, the identification of specific drug(s) with the ability to reduce the incidence of a particular cancer type in selected groups of patients offers a promising strategy to reduce cancer metastasis for that cancer type. This notion is supported by a recent study that tested the effect of digoxin, a Na^+^/K^+^ ATPase inhibitor typically prescribed for cardiovascular conditions. The authors found that digoxin suppress circulating tumor cell clusters and blocked metastasis in breast cancer patients treated daily for one week with a maintenance digoxin dose (0.7–1.4 ng/mL = 8.96–1.79 nM serum level) [226]. For comparison, the IC_50_ of digoxin for different breast cancer cell lines was 60 nM, 230 nM, 80 nM and 170 nM for MCF-7, BT-474, MDA-MB-231 and ZR-75–1 breast cancer cell lines, respectively [227]. Pathway analysis in samples collected and processed for next-generation RNA-seq showed highly significant downregulation of cell-cycle-related genes [226]. Tamoxifen is a particular drug that adds clinical evidence for the PLDA with the aim of preventing metastasis. Tamoxifen is an anti-estrogenic substance effective in the adjuvant therapy applied in human breast cancer. When prescribed to women with estrogen receptor (ER)-positive early breast cancer for 5–10 years, it reduced the risk of breast cancer recurrence, reduced breast cancer mortality, and reduced overall mortality [228]. The anticancer effect of tamoxifen is believed to be due to the hydroxylated metabolites, 4-hydroxytamoxifen (4OHtam), and 4-hydroxy-N-desmethyltamoxifen (4OHNDtam/endoxifen), because of their high affinity for the ER [229]. Its main active metabolite, 4-hydroxytamoxifen, has an anticancer effect when tested in vitro with an IC50 between 18 and 27 μM for MCF-7 and MDA-MB-231 human breast cancer cell lines, respectively. These concentrations are higher than the plasma concentration that ranges between 0.0213 and 0.0227 μM [230]. However, the lowest effective in vitro concentrations of endoxifen (20–20 nM) are within the plasma concentration range (5–80 nM) found in patients (see Table 4). Thus, tamoxifen provides a good example of a drug that when used for prolonged time, at relative low concentrations, prevents metastasis not directly but possibly through one of its active metabolites. This raises the possibility that some of the FDA-approved drugs listed in Table 1 are prodrugs and a few of their unknown or known but not well-characterized active metabolites target key biological processes in cancer and non-cancer cells (as recently reported for aspirin) that drive tumor relapse and metastasis.

Providing that initiation of both the primary tumor and metastasis in the same type of cancer (in the same patient) share the same mechanisms, it would be possible to use PLDA of a known drug to reduce metastasis formation. This concept is illustrated in Figure 1 using a hypothetical drug X in a specific subtype of triple-negative breast cancers carrying a specific mutation. This strategy can be applied to other subtypes of cancers. It can be anticipated that prevention of metastasis will require the identification of several specific drugs, each of which targets a specific cancer subtype, leading to a new approach to personalized medicine in oncology.

## Figures and Tables

**Figure 1 ijms-26-02720-f001:**
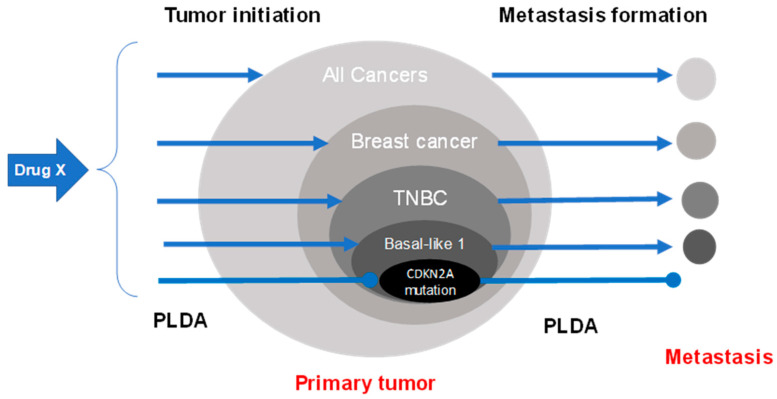
Potential use of PLDA of FDA-approved drugs to reduce the metastasis formation of specific cancer subtypes in selected populations. In this example, a hypothetical drug X known to reduce the incidence (carcinogenesis) of a specific subtype of triple-negative breast cancer (TNBC, basal-like subtype, with mutation in the CDKN2A gene) may be useful to prevent metastasis in patients carrying this cancer subtype.

**Table 1 ijms-26-02720-t001:** Partial list of FDA-approved drugs (except genistein and resveratrol, not FDA approved) with known effects on reducing (indicated by down arrows) cancer incidence.

Drug	Indication	Cancer Type	References
Metformin	DBT type II	ProstateC ↓, ColorectalC ↓, BreastC ↓	[3,4,5]
Aspirin	Analgesic, antipyretic and agent for cardiovascular prophylaxis	ColorectalC ↓, PancreaticC ↓, Ovarian C ↓	[6,7,8]
Statin (simvastatin, atorvastatin, pravastatin, fluvastatin, rosuvastatin, and pitavastatin)	LDL-cholesterol, type 2 diabetes mellitus (T2DM)	ProstateC, BreastC, LungC, ColorectalC ↓	[11,12]
Simvastatin	Lowers cholesterol (T2DM)	Renal cell carcinoma ↓	[9,13]
Glipizide	Type 2 diabetes mellitus (T2DM)	ProstateC ↓	[14]
Glimepiride-metformin	Type 2 diabetes mellitus (T2DM)	BreastC ↓	[15,16]
Empagliflozin	Type 2 diabetes mellitus (T2DM)	BladderC ↓	[16,17]
Naproxen	Analgesic, antipyretic, and anti-inflammatory drug	Urinary BladderC ↓, BreastC ↓	[18,19]
Etoricoxib	Non-steroidal anti-inflammatory drug	Colon C ↓	[20]
Everolimus	Organ transplantation; new pediatric dosage used to treat subependymal giant cell astrocytoma (SEGA)	BreastC ↓	[21,22]
Exemestane	Estrogen modulator	BreastC ↓	[23]
Goserelin (Zoladez)	Reduction in plasma/serum estrogen levels in pre- or perimenopausal women	BreastC ↓	[24]
Aldesleukin	Immunotherapy drug	Renal cell carcinoma (RCC) or KidneyC ↓	[25]
Raloxifene	Estrogen receptor modulator, bone health	Breast C ↓	[26]
Lenalidomide	Refractory prurigo nodularis	Multiple myeloma ↓	[27,28]
Phenformin	Anti-diabetic agent, phenethylbiguanide	Ovarian C, Breast C ↓	[29,30]
Tretinoin	Anti-inflammatory properties; acne	Acute promyelocytic leukemia ↓	[31,32]
Degarelix (Firmagon)	Gonadotropin-releasing hormone receptor antagonist (hormone therapy drug)	Hormone-dependent prostate C ↓	[33]
Resveratrol	Plant compound that acts against pathogens, mostly found in red grapes and products made from those grapes (wine)	Prostate C ↓, Colon C ↓, Breast C ↓	[34,35,36]
Genistein	Protein tyrosine kinase and topoisomerase II inhibitor, present in soy	Breast C ↓, Prostate C ↓	[37,38,39,40]

**Table 4 ijms-26-02720-t004:** Typical in vitro versus in vivo concentrations of tamoxifen.

Drug	Typical Duration of Treatment	Typical Plasma/Serum Concentration	Typical In Vitro Concentration	Lowest In Vitro Effectiveness Within Plasma Concentration Range
Tamoxifen	5–10 years [228]	124–133 ng/mL = 0.33–0.36 μM [231]	2.230 μM for the MDA line MB 231, 10.045 μM for the MCF7 line and 4.579 μM for the HCC 1937 line [232]	NO
4-hydroxytamoxifen	Same as tamoxifen	8.26–8.80 ng/mL = 0.0213–0.0227 μM [231]	27 μM (IC50 for MCF-7) 18 μM IC50 for MDA-MB 231) [230]	NO
4-hydroxy-N-desmethyltamoxifen (4OHNDtam, endoxifen)	Same as tamoxifen	5–80 nM in the serum of tamoxifen treated patients [233]	low concentrations 20–40 nM significantly repress the estrogen-induced growth of MCF7 cells high concentrations (100–1000 nM) either completely block or drastically repress this response [234]	YES

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
