# Peer review of "Prolonged Low-Dose Administration of FDA-Approved Drugs for Non-Cancer Conditions: A Review of Potential Targets in Cancer Cells"

_ijms, 2025, doi:10.3390/ijms26062720_

Round 1
Reviewer 1 Report
Comments and Suggestions for Authors
In this manuscript, the authors did excellent summaries on the most used FDA approved drugs for non-cancer conditions and proposed some mechanism of actions which highlighted new directions for anti-cancer research. Overall, this manuscript is well-organized and some concepts worth further investigation. I have the following suggestions.
- It would be great if the authors could also summarize some popular anti-cancer medications in vitro/in vivo comparison like Table 2. BTY, there is an empty line in Table 2 after Goserelin.
- For the in vitro experiments, the cell lines might be different from the cancer cells in vivo, either in environments or genetic components or other factors. Maybe the authors can comment on that.
Author Response
Reviewer #1 comments
In this manuscript, the authors did excellent summaries on the most used FDA approved drugs for non-cancer conditions and proposed some mechanism of actions which highlighted new directions for anti-cancer research. Overall, this manuscript is well-organized and some concepts worth further investigation. I have the following suggestions.
- It would be great if the authors could also summarize some popular anti-cancer medications in vitro/in vivo comparison like Table 2. BTY, there is an empty line in Table 2 after Goserelin.
Response: In this revised version we summarized a few popular anticancer drugs (Doxorubicin and Cyclophosphamide ).
The empty line in Table 2 was removed.
A few popular anticancer drugs were summarized in Section 3 and Table 3 (see pages 5-7)
- For the in vitro experiments, the cell lines might be different from the cancer cells in vivo, either in environments or genetic components or other factors. Maybe the authors can comment on that.
Response: the following paragraph was added in (section XX) to address this reviewer concern.
“Moreover, while the cell lines used in vitro experiments are mainly from commercial sources that might be different from the cancer cells in vivo this does not explain the general discrepancy. This assumption is supported by the fact that when drugs are tested in commercial cell lines and patient derived cell lines the overall potency is similar. For instance, we found that commercial DBTRG.05MG glioma cells showed similar sensitivity to Menadione and Vitamin C compared to a panel of eight different glioma patient-derived cell lines. (Page 3)
Reviewer 2 Report
Comments and Suggestions for Authors
This review focuses on FDA-approved drugs for non-cancer diseases. It explores the potential effects of long-term, low-dose administration of several commonly used FDA-approved non-cancer drugs on cancer cells. Although these drugs were not originally designed for cancer treatment, they have been found to reduce cancer incidence. Additionally, the paper highlights the discrepancy between the in vitro drug concentrations required to eliminate cancer cells and the plasma concentrations tolerated by the human body, for which no unified explanation currently exists. Furthermore, it discusses potential mechanisms such as cancer stem cells and cellular senescence in cancer progression, proposing that long-term low-dose administration may target cancer cells through multiple pathways. However, the evidence remains insufficient, and future research may refine chemoprevention strategies and contribute to personalized cancer therapy.
Major Comments:
- Enhancing the Introduction:
The introduction could include more examples of FDA-approved non-cancer drugs that have been used in cancer treatment to strengthen the argument. Additionally, it is recommended that the authors clearly state the novelty of this review compared to existing literature and provide a more structured summary of the article's content at the end of the introduction.
- Pharmacokinetic Considerations:
The second section discusses the discrepancy between drug plasma concentrations found in patients and the in vitro concentrations required to eliminate cancer cells. However, it lacks a discussion on pharmacokinetic factors, leading to an incomplete logical framework. The authors are advised to supplement this section with relevant pharmacokinetic insights.
- Improving Section Headings:
Some section headings are too broad and do not accurately summarize the content of the corresponding paragraphs. It is recommended that the authors incorporate more descriptive terms to better guide readers in understanding the main points of each section.
- Strengthening the Conclusion:
While the conclusion addresses the current research limitations and future directions, it does not comprehensively discuss the constraints. For instance, different types of cancer may only be preventable by specific drugs, and providing more concrete examples would improve clarity. Additionally, the proposed future research directions should be supported by specific literature and the latest studies. Strengthening this section with more references and up-to-date research findings is highly recommended.
Comments on the Quality of English LanguageThis review focuses on FDA-approved drugs for non-cancer diseases. It explores the potential effects of long-term, low-dose administration of several commonly used FDA-approved non-cancer drugs on cancer cells. Although these drugs were not originally designed for cancer treatment, they have been found to reduce cancer incidence. Additionally, the paper highlights the discrepancy between the in vitro drug concentrations required to eliminate cancer cells and the plasma concentrations tolerated by the human body, for which no unified explanation currently exists. Furthermore, it discusses potential mechanisms such as cancer stem cells and cellular senescence in cancer progression, proposing that long-term low-dose administration may target cancer cells through multiple pathways. However, the evidence remains insufficient, and future research may refine chemoprevention strategies and contribute to personalized cancer therapy.
Major Comments:
- Enhancing the Introduction:
The introduction could include more examples of FDA-approved non-cancer drugs that have been used in cancer treatment to strengthen the argument. Additionally, it is recommended that the authors clearly state the novelty of this review compared to existing literature and provide a more structured summary of the article's content at the end of the introduction.
- Pharmacokinetic Considerations:
The second section discusses the discrepancy between drug plasma concentrations found in patients and the in vitro concentrations required to eliminate cancer cells. However, it lacks a discussion on pharmacokinetic factors, leading to an incomplete logical framework. The authors are advised to supplement this section with relevant pharmacokinetic insights.
- Improving Section Headings:
Some section headings are too broad and do not accurately summarize the content of the corresponding paragraphs. It is recommended that the authors incorporate more descriptive terms to better guide readers in understanding the main points of each section.
- Strengthening the Conclusion:
While the conclusion addresses the current research limitations and future directions, it does not comprehensively discuss the constraints. For instance, different types of cancer may only be preventable by specific drugs, and providing more concrete examples would improve clarity. Additionally, the proposed future research directions should be supported by specific literature and the latest studies. Strengthening this section with more references and up-to-date research findings is highly recommended.
Author Response
Reviewer #2 comments
This review focuses on FDA-approved drugs for non-cancer diseases. It explores the potential effects of long-term, low-dose administration of several commonly used FDA-approved non-cancer drugs on cancer cells. Although these drugs were not originally designed for cancer treatment, they have been found to reduce cancer incidence. Additionally, the paper highlights the discrepancy between the in vitro drug concentrations required to eliminate cancer cells and the plasma concentrations tolerated by the human body, for which no unified explanation currently exists. Furthermore, it discusses potential mechanisms such as cancer stem cells and cellular senescence in cancer progression, proposing that long-term low-dose administration may target cancer cells through multiple pathways. However, the evidence remains insufficient, and future research may refine chemoprevention strategies and contribute to personalized cancer therapy.
Major Comments:
- Enhancing the Introduction:
The introduction could include more examples of FDA-approved non-cancer drugs that have been used in cancer treatment to strengthen the argument.
Response: this review is not about “FDA-approved non-cancer drugs that have been used in cancer treatment” but about no anticancer FDA-approved drugs that have been prescribed for non-cancer conditions that shows benefit by preventing cancer”.
Additionally, it is recommended that the authors clearly state the novelty of this review compared to existing literature and provide a more structured summary of the article's content at the end of the introduction.
Response: To explain the novelty of this review and provide a more structured summary of the article we added the following paragraph: “. The goal of this article is to review some of the most used FDA approved drugs prescribed for non-cancer conditions, highlighting the discrepancies between in vivo vs in vitro potency as anticancer drugs and their possible mechanism of action as chemopreventive agents. We will discuss the existing evidence that link these effects on these drugs on CSCs, cellular senescence, clonogenicity as well as other potential targets. “
- Pharmacokinetic Considerations:
The second section discusses the discrepancy between drug plasma concentrations found in patients and the in vitro concentrations required to eliminate cancer cells. However, it lacks a discussion on pharmacokinetic factors, leading to an incomplete logical framework. The authors are advised to supplement this section with relevant pharmacokinetic insights.
Response: We agree with the reviewer that pharmacokinetic factors found in patients are important. To address this concern we added the following paragraph: “The discrepancies cannot be explained by pharmacokinetics factors present in vivo but absent in vitro experiments. For instance, in in vitro experiments there are no metabolic elimination and clearance drugs by other organs such as liver and kidneys. Thus, it would be expected that if pharmacokinetics factors play a role, these drugs will be more potent in vitro than in vivo. (Page 3) .
We also added relevant information of active metabolites of a few anticancer (Cyclophosphamide and Capecitabine, Tamoxifen), and non-anticancer (aspirin) FDA approved drugs (See table 2, 3 and 4 as well as Sections 3 and 8)
- Improving Section Headings:
Some section headings are too broad and do not accurately summarize the content of the corresponding paragraphs. It is recommended that the authors incorporate more descriptive terms to better guide readers in understanding the main points of each section.
Response: Section headings have been modified to reflect the content.
- Strengthening the Conclusion:
While the conclusion addresses the current research limitations and future directions, it does not comprehensively discuss the constraints.
For instance, different types of cancer may only be preventable by specific drugs, and providing more concrete examples would improve clarity.
Response: This concern was addressed in the original manuscript. “On the other hand, the available data shown in Table 1 indicates that there is no universal drug that can lower the incidence of all types of cancer. This notion is consistent with the paradigm that all existing anti-cancer drugs are cancer type-specific, targeting different mechanisms involved in the development and progression of tumors. Hence, each type of cancer may be prevented only by a few specific drugs. For instance, metformin reduces the incidence of prostate, colorectal and breast but has not known effect on, for example, renal cell carcinoma or pancreatic cancer. This seems to be the trend for all FDA approved drugs known to reduce cancer incidence. It is also important to clarify that metformin does not prevent 100% but only reduces the incidence of, let say, breast cancer. Thus, even for a particular type of cancers, only a fraction of patients will benefit by taking metformin, so, the likelihood of eliminating cancer incidence with a single “magic” chemopreventive agent is remote.
Additionally, the proposed future research directions should be supported by specific literature and the latest studies. Strengthening this section with more references and up-to-date research findings is highly recommended.
Response: This concern was addressed in the original manuscript “However, the identification of specific drug(s) with the ability to reduce the incidence of a particular cancer type in selected groups of patients offers a promising strategy to reduce cancer metastasis for that cancer type. Providing that initiation of both the primary tumor and metastasis in the same type of cancer (in the same patient) share the same mechanisms, it would be possible to use PLDA of a known drug to reduce metastasis formation. This concept is illustrated in Figure 1 using a hypothetical drug X in a specific subtype of triple negative breast cancers carrying a specific mutation. This strategy can be applied to other subtypes of cancers. It can be anticipated that prevention of metastasis will require the identification of several specific drug each of which targeting a specific cancer subtype leading to a new approach to personalized medicine in oncology. “
In this revised version, we added additional information related to two drugs (Digoxin and Tamoxifen) to strengthen this section.
Round 2
Reviewer 2 Report
Comments and Suggestions for Authors
The authors have responded to the reviewers' concerns and have no further comments